# Dual-Labelling Strategies for Nuclear and Fluorescence Molecular Imaging: Current Status and Future Perspectives

**DOI:** 10.3390/ph15040432

**Published:** 2022-03-31

**Authors:** Manja Kubeil, Irma Ivette Santana Martínez, Michael Bachmann, Klaus Kopka, Kellie L. Tuck, Holger Stephan

**Affiliations:** 1Helmholtz-Zentrum Dresden-Rossendorf, Institute of Radiopharmaceutical Cancer Research, Bautzner Landstraße 400, 01328 Dresden, Germany; m.kubeil@hzdr.de (M.K.); i.santana-martinez@hzdr.de (I.I.S.M.); m.bachmann@hzdr.de (M.B.); k.kopka@hzdr.de (K.K.); 2National Center for Tumor Diseases (NCT) Dresden, University Hospital Carl Gustav Carus, Fetscherstraße 74, 01307 Dresden, Germany; 3Faculty of Chemistry and Food Chemistry, School of Science, Technische Universität Dresden, 01069 Dresden, Germany; 4School of Chemistry, Monash University, Melbourne, VIC 3800, Australia

**Keywords:** molecular imaging, positron emission tomography, single-photon emission computed tomography, near-infrared fluorescence

## Abstract

Molecular imaging offers the possibility to investigate biological and biochemical processes non-invasively and to obtain information on both anatomy and dysfunctions. Based on the data obtained, a fundamental understanding of various disease processes can be derived and treatment strategies can be planned. In this context, methods that combine several modalities in one probe are increasingly being used. Due to the comparably high sensitivity and provided complementary information, the combination of nuclear and optical probes has taken on a special significance. In this review article, dual-labelled systems for bimodal nuclear and optical imaging based on both modular ligands and nanomaterials are discussed. Particular attention is paid to radiometal-labelled molecules for single-photon emission computed tomography (SPECT) and positron emission tomography (PET) and metal complexes combined with fluorescent dyes for optical imaging. The clinical potential of such probes, especially for fluorescence-guided surgery, is assessed.

## 1. Introduction

Over the past 20 years, molecular imaging has proved a valuable technique for visualisation and characterisation of pathophysiological processes in general but especially in the field of cancer research [1,2,3,4,5]. Nuclear techniques such as single photon emission computed tomography (SPECT) and positron emission tomography (PET) play a special role here because they have an extremely high sensitivity (the nanomolar and even the picomolar level), an almost unlimited penetration depth in biological tissues and they provide quantitative data on the pharmacokinetic properties of radiolabelled drugs [6,7,8,9,10]. PET has advantages over SPECT, particularly with regard to spatial resolution and quantification. This is due to the different distribution of the emitting photons. In SPECT, the gamma quanta are distributed statistically and in a rather disorderly way, whereas PET uses the defined collinear emission and detection of two 511 keV gamma photons, which are formed during the annihilation process of electrons and positrons (Figure 1). Commonly used gamma-emitting radionuclides for SPECT are ^99m^Tc, ^111^In and ^123^I, while for PET mostly the positron-emitting radionuclides ^18^F, ^64^Cu, ^68^Ga, ^89^Zr and ^124^I are applied. For radiolabelling of longer circulating objects such as natural antibodies or nanomaterials, radionuclides with a rather longer half-life are used to follow the pharmacokinetic properties over several days. Increasingly, theranostic radionuclides like lutetium-177 (SPECT and β-emitter) as well theranostic pairs like scandium-44 (PET)/scandium-47 (β-emitter), copper-64 (PET)/copper-67 (β-emitter), strontium-83 (PET)/strontium-89 (β-emitter), yttrium-86 (PET)/yttrium-90 (β-emitter), iodine-124 (PET)/iodine-131 (β-emitter) and terbium-152 (PET)/terbium-161 (β-emitter), which allow both monitoring and treatment, are being used or are under discussion [11]. In total, more than 300 radionuclides are available for use in medicine, biology, chemistry and other fields [12].

This review also focusses on optical imaging (OI) in the context of fluorescence, phosphorescence or luminescence spectroscopy. Optical imaging has about one order of magnitude lower sensitivity to nuclear techniques with superior resolution in the micro-meter range (Figure 2A) [13,14,15]. However, it suffers from the drawback of limited penetration depth, which means that application in the clinic can be limited. This limitation has been overcome to some degree by continual improvements of fibre-optic endoscopic probes, thus allowing higher image penetrating depths. However, the real advancement has been continual development of near-infrared (NIR) probes, further detailed below, which has allowed the depth of study to be increased from ~0.5 mm (400 nm) to up to detection of several centimetres at the excitation wavelengths greater than 1500 nm. This is due to two phenomena: (i) tissue penetration is wavelength dependent; thus, the greater the wavelength, the deeper the penetration and (ii) water, which is prevalent in tissues, absorbs IR light which has the effect of greater depth and greater contrast (Figure 2B) [16].

Dyes that are excited by NIR irradiation are classified as NIR-I (650–900 nm range), NIR-IIa (1000–1400 nm) and NIR-IIb (1500–1700 nm) probes. Indocyanine green and methylene blue are two small molecule NIR-I dyes that are approved for use in the clinic. These dyes, while readily excreted, are non-specific; furthermore, as NIR-I dyes they suffer from the typical photon scattering and poor photon absorption. The BODIPY scaffold (Figure 3) can be further functionalised to increase absorption and emission wavelength. The second near-infrared (NIR-II) window, comprised of probes containing inorganic or organic fluorophores, are superior due to the lower light scattering, the higher maximum permissible exposure that can be used and the greater image penetrating depth. The chemical structures of NIR dyes that are typically used for modular ligand and nanoscale systems, as well as the chemical structures relevant to fluorescence molecular imaging described in the subsequent sections, are reproduced in Figure 3. The synthesis of molecules containing donor–acceptor–donor (D-A-D) has allowed organic molecules to emit in the NIR-II window; however, the synthesis of these molecules is often challenging and as such it has not been widely performed for in vivo imaging. There are a number of good, recent reviews summarising NIR-I and NIR-II dyes [13,20,21,22,23,24] and we refer the reader to them for a more comprehensive understanding of their design and use. The recent developments of NIR-II metal-based luminescent complexes are the focus of this section and thus detailed below. Inorganic NIR-II fluorophores such as carbon nanotubes or AuNP are outside the scope of this review.

The nuclear and optical dual-labelled imaging agents reported so far are mainly based on the combination of radiolabelled compounds that use the radionuclides fluorine-18, copper-64, gallium-68, zirconium-89, technetium-99 m and indium-111 and organic dyes. Such dual modality probes are summarised in a series of reviews documenting the rapid development of this exciting field [30,31,32,33,34,35,36,37]. The two recent reviews look at the use of these samples from a clinical perspective [37] and discuss the multiple modification possibilities through the use of different conjugation strategies [30]. In the latter review, an extensive compilation of nuclear and fluorescence imaging molecular and nanoscale tools with a discussion of the pros and cons for special biological applications can be found.

The pharmacokinetic requirements of dual-labelled imaging agents in vivo are dependent on the mode of delivery as well as the timeframe between the administration and imaging test. Furthermore, the type of chelators, dye molecules and target modules, as well as their chemical linkage, will make these properties vary and so comprehensive pre-clinical studies are needed for further developments. Some of the examples discussed below utilise ligands or fluorophores that have been approved for use in the clinic. However, a comprehensive understanding of the agent’s absorption, distribution, metabolism and excretion (ADME) properties, as well as toxicity, is naturally needed before phase III trials can commence.

In our review, we focus in particular on novel molecular systems for clinical use in cancer medicine, with an emphasis on new modular ligands and nanoscale systems.

## 2. Fluorescence Imaging for Biomedical Applications

### 2.1. NIR Metal Complex Imaging Agents

It is only recently, with the key advances discussed below, that NIR metal complexes for whole body imaging can be achieved, rather than using cell imaging alone. The findings below are highlights of the field with a focus on the use of NIR metal complexes in whole body imaging studies and the key findings that have accumulated in the discovery of such complexes, rather than the broader area of time-resolved lanthanide imaging.

#### 2.1.1. Lanthanide-Based Molecules

The fascinating and unique optical properties of lanthanide(III) complexes have intrigued scientists for decades and their potential use as bioprobes was noted as early as the 1970s [38]. Their photoluminescent properties are a consequence of their [Xe]4f^n^ electronic configuration, with the 4f–4f transitions resulting in spectra in the visible to NIR region. However, lanthanide ions themselves are weakly absorbing due to their small molar absorption coefficients (<10 M^−1^·cm^−1^), which is a consequence of Laporte forbidden 4f transitions [39]. At the same time, the resulting long-lived luminescence, due to the Laporte forbidden 4f–4f transition of metal electrons, is a highly attractive property, to obtain it the “antenna effect”, first coined by Weissman in 1942, needs to be exploited [40]. In this case, a highly absorbing ligand, often organic in nature, and whose triplet energy state is at the appropriate level for transfer to the lanthanide excited state by intermolecular energy transfer, needs to be covalently attached or close in space to the receiving lanthanide(III) ion [41]. Typically this is accomplished by functionalisation of the multidentate ligand, to which the lanthanide(III) ion is complexed, with the appropriate antenna moiety. Ligands with negatively charged or neutral oxygen and nitrogen donor atoms give highly stable complexes; see Figure 10 for the chemical structures of representative chelators. DOTA and DO3A, or their derivatives, result in the most stable complexes (log *K* = 23–25) [42,43].

To date, the luminescent imaging in vitro of terbium(III) and europium(III) complexes has been well explored; however, lanthanide(III) based emitters in the NIR are more scarce [44,45]. The incorporation of ligands that absorb in the NIR region, as well as the two-photon (2P) absorption, have allowed lanthanide(III) complexes to be used for optical imaging [46]. The next section will highlight recent key findings in this area.

The 8-coordinate cationic [YbL]^+^ complex (Figure 4A) was utilised for 2P-imaging of living cells, with excitation wavelength 800 nm. This, the first reported Yb^III^ 2P-luminescent probe, was the result of a decade of research in which the antenna, chelate and potential for 2P-bioimaging were optimised. In addition to the inclusion of the 2P-antenna, the cationic complex over a neutral one ensured that cell internalisation occurred readily [47].

Following this report, a number of Yb^III^ porphyrinate complexes (Yb-4, Yb-2 and Yb-*cis*/*trans*-3, Figure 4B) were disclosed for 1P- and 2P-imaging [48,49,50]. The porphyrinates typically have intense bands at approximately 620 nm, suitable for Yb^III^ excitation, and large extinction coefficients. The first report (2018) noted that substitution of meso-phenyl groups can modify Yb-NIR emission. Use of the deuterated Kläui ligand allowed β-fluorinated and non β-fluorinated complexes to be compared. Complex Yb-4 gave the most favourable properties and was further investigated for NIR imaging (excitation 620 nm; emission 935 nm; quantum yield 10%). Due to the long luminescent lifetimes of Yb^III^ complexes, in vitro confocal time-resolved fluorescence lifetime imaging (FLIM) allowed for the removal of cell autofluorescence. In this ground-breaking work, the authors note that porphyrinoid ligands are exciting prospective candidates for NIR molecular probes [50]. In two follow up reports, the authors extend this concept and utilise the molecular probes for in vivo NIR-II imaging. The probes investigated have quantum yields of about 10% in water and probe Yb-2, a water-soluble carboxylate, was further investigated due to its highest resolution and signal-to-background ratio properties. When excited at 532 nm, detection of the NIR-II luminescence signals at a depth of 8 mm in a tissue sample could be observed. In vivo NIR-II fluorescence imaging showed the potential for this probe to be used for bioimaging [48]. The modification of the porphyrinoid Yb-3 resulted in regioisomers with differing properties; the *cis* isomer was suitable for NIR-II imaging whereas the *trans* isomer, upon irradiation, produced singlet oxygen [49].

In a very recent report (2021), the photophysical properties of a range of lanthanide-based carbazole-containing porphyrinoid complexes (Figure 4B, Ln-L, Ln = Gd, Yb and Er) have been further modified and examined in vitro and in vivo. As above, the coordinating ligand, a carbazole-based porphyrinoid, was chosen due to an intense absorption band at 630 nm and a large extinction coefficient. The complexes were investigated for their potential usage as photothermal therapeutics as well as NIR imaging agents. The lanthanide complexes exhibited a NIR absorption at 706 nm with the Yb^III^ complex yielding the most encouraging results in vitro. For the encapsulation of the Yb^III^ complex in mesoporous silica nanoparticles after intravenous injection, the in vivo studies confirmed that the photoirradiation of the tumour using a NIR laser (690 nm), with temperature monitoring, could be used to monitor tumour progress [51].

A number of pyclen-based ligands have been explored in the development of a family of lanthanide-based luminescent probes (Figure 5). The findings build on previous work within the group and others, where the photophysical different chelates and lanthanide ions were investigated for bioimaging applications. In this report, the lanthanide complexes, in all cases, have a coordination number of 9, thereby resulting in hydration numbers (q) of 0 or below 1. The lanthanide complexes, [EuL^4a^], [SmL^4a^], [YbL^4b^], [TbL^4c^], [DyL^4c^] and [EuL^4a’^] (Figure 5), can be excited between 300 and 400 nm. Depending on the lanthanide complex, they can also undergo 2P-excitation (excitation between 700 and 900 nm), which is more valuable for in vivo bioimaging applications. The results from the in vitro cellular studies are shown in Figure 5. Further studies with [EuL^4a^] in zebrafish embryos, which was shown to be non-toxic, and 2P-excitation resulted in a high-resolution image. The authors highlight the potential of these lanthanide-based luminescent probes for imaging thick tissue and subsequent diagnosis of disease [52]. The ^161^Tb and ^177^Lu complexes of these ligands are thermodynamically stable and kinetically inert. Thus, such ligand complexes have the potential for radionuclide therapy as well as imaging [53].

Recently, in the development of lanthanide-based nanocomposites for cancer therapy, a nanocomposite composed of DOTA as the chelate and camptothecin as the toxic payload (cycLN-ss-CPT, Ln = Gd^III^ or Yb^III^, Figure 6) has been utilised [54]. In this study, the Ln^III^ ratio was controlled via precise chemical synthesis of the Gd^III^ and Yb^III^ complexes, and upon formation of the micellar LnNP and excitation at 330 nm, the typical Yb^III^ emission spectrum was observed. Incubation of the Gd/YbNPs in HeLa cells confirmed, via NIR optical imaging, that such nanocomposites could be used to monitor uptake. Gd^III^ was included for in vivo MR imaging.

Ligand complexes of Eu^III^ and Tb^III^ alone are not able to be used for in vivo optical imaging, as the efficient energy transfer to these lanthanide ions typically requires external excitation in the region of 250−350 nm. Recently, it was communicated that careful design of the complexes can allow for in situ excitation via Cerenkov radiation (CR) (Figure 7). In this example, the administration of radiofluorine (Na^18^F) with the lanthanide complex allowed for optical and multiplex imaging concurrently [55].

#### 2.1.2. Non-Lanthanide-Based Molecules

Luminescent iridium complexes, due their excellent photo-stability and high quantum yields, have been utilised as intracellular sensors especially for detection of oxygen, reactive oxygen species (ROS) and other endogenous species [56]. A number of recent NIR-emitting iridium complexes that can be used for in vivo imaging are reproduced in Figure 8. The iridium(III) cyanine complex nanoparticles IrCy-NPs allowed NIR absorption and singlet oxygen generation upon irradiation at 808 nm [57]. The iridium(III) complex-derived polymeric micelle PolyIrLa (the conjugated iridium(III) complex with UNCPs) allowed photodynamic therapy and chemotherapy to occur (NIR irradiation at 980 nm) [58]. The iridium(III) complex IrDAD, containing a donor–acceptor–donor (D-A-D) moiety, allowed for the formulation of a nanoparticulate system (IrDAD-NPs) that can be used for NIR-dual imaging and phototherapy. Tissue penetration was observed and NIR irradiation (808 nm) resulted in the formation of ROS and heat [59].

Typically, ruthenium(III) complexes emit in the visible region and as a result until recently have not been used for imaging studies. The Ru(II) polypyridyl complex, HL-PEG_2K_ (Figure 9), constructed using the D-A-D strategy of an organic NIR-II fluorophore H_4_–PEG-Glu [60], allowed NIR-II imaging and chemo-photothermal therapy to occur simultaneously. Interestingly, in vivo studies revealed that HL-PEG_2K_, when compared to *cis*platin, had lower toxicity and better activity [61].

Section 3 describes the use of optical dyes as part of the multifunctional ligand systems for nuclear and optical dual imaging and Section 4 outlines their incorporation in nanoscale systems.

## 3. Modular Ligand Systems

Multimodal imaging based on nuclear and fluorescence probes allows for synergy of these modalities. The goals are improved non-invasive visualisation and quantification of the underlying processes (occurring at the molecular level), tumour localisation and the possibility of image-guided surgery. For these purposes, it is necessary to design sophisticated bimodal imaging probes that satisfy the demands of more than one imaging modality within a small molecule or a nanoscale system (vide infra).

Frequently, low molecular weight compounds are involved in the design of probes for bimodal imaging. These compounds enable the assembly of moieties suitable for the desired imaging channels. Such moieties include fluorescent dyes for optical imaging, leaving groups suitable to introduce PET/SPECT radionuclides or bifunctional chelator agents (BFCAs) for labelling with radiometals. In particular, there is a need for molecules that allow the simple introduction of fluorophores, radionuclides and targeting modules at the same time.

In recent years, several multimodal imaging ligands have been studied. Thus, the library of options for the development of suitable dual tools, whilst comprehensive, is still expanding. Some of the most representative systems, which include multifunctional organic systems, frequently used bifunctional chelating agents (BFCAs) and newly developed modular ligand systems, are discussed below (Figure 10).

### 3.1. Organic-Based Systems

Organic modular systems use suitable leaving groups or isotope exchange for the introduction of non-metallic radionuclides (^18^F, ^11^C, ^123^I, ^124^I, etc.) using covalent chemical bonds. Furthermore, these systems have additional functional groups that allow the incorporation of fluorescence labels and targeting vector molecules. Some recently reported examples are presented below.

The frequently used PET radionuclide ^18^F has been studied extensively for the radiolabelling of organic dyes. BODIPY [62,63,64], rhodamines [65], xanthene derivatives [66] and cyanines [67] are among the most common radiofluorinated fluorophores. These dyes can be tailored with diverse leaving groups (or undergo isotopic exchange) for the introduction of radioisotopes such as ^18^F. This strategy has been exploited to achieve improved radiochemical yields, easier synthesis and more effective purification methods. In this context, in 2019, Kim et al. reported an ^18^F-labelled BODIPY dye, suitable for PET/Optical imaging [64]. The radiofluorination proceeded through an isotopic exchange (^19^F-^18^F), mediated by the Lewis acid SnCl_4_. Quantitative radiochemical yield (RCY) and high molar activity were achieved. More importantly, the radiofluorinated dye showed favourable pharmacokinetics and allowed for the simultaneous application of PET and optical imaging (OI) of the brain. More recently, the group Kopka et al. studied radiolabelled silicon-rhodamines (SiRs) [68]. The yielded SiRs display distinctive near-infrared (NIR) optical properties, large quantum yields and high photo-stability. Furthermore, the boronic acid (leaving moiety) enabled the introduction of ^18^F and ^123^I (using SiRs as a common precursor). Radiolabelling with high molecular activities was achieved using copper-mediated radiofluorination and copper-mediated radioiodination, respectively. These radiofluorinated molecules are suitable for co-localisation experiments (assessed by fluorescence confocal microscopy). Overall, the developed ligand structure allows for the simultaneous application of PET or SPECT and NIR imaging. Radiofluorinated dye molecules often have a lipophilic character and thus are especially beneficial for clinical applications in imaging of the brain but also of the heart.

### 3.2. Metal-Based Systems

Metal-based systems use BFCAs for the stable binding of metallic radionuclides. Of particular importance in the design of BFCAs is the high complex stability and kinetic inertness as well as the use of appropriate functional groups for direct or linker-mediated conjugation with fluorescent dyes and/or biomolecules. In recent years, various BFCAs have been developed for SPECT/OI and PET/OI, using common SPECT (^111^In, ^99m^Tc) and PET (^64^Cu, ^68^Ga, ^89^Zr) radionuclides.

One of the first examples of targeted SPECT/OI imaging was presented by Wang et al., who described a dual-labelled agent for imaging the interleukin-11 receptor (IL-11Rα) [69]. The dual-labelled probe consisted of a peptide (targeting IL-11Rα) conjugated to ^111^In-DTPA and the fluorophore IR-783. The conjugate allowed for the clear visualisation of the ligand-antigen interaction in tumour-bearing mice. This report has served as a basis for further research of imaging IL-11Rα-expressing lesions with further fluorophores, such as Cy7 [70]. Very recently, a dual-labelled prostate-specific membrane antigen (PSMA)-targeted probe was developed using the dye IRDye800CW for NIR imaging and SPECT with ^111^In-DOTA or ^99m^Tc-MAG3 (mercaptoacetylglycylglycylglycine). The high-affinity ligand, consisting of naphthylalanine, aminomethyl benzoic, glutamic and nicotinic acid, allowed for efficient in vivo imaging of PSMA-expressing tumours (Figure 11). The pharmacokinetic properties of the dual samples can be adjusted both by the choice of ligand and by the conjugation chemistry used.

Since the first report in 2011, the potential of heterobimetallic ^99m^Tc/Re complexes for bimodal SPECT/fluorescence imaging has been studied [72]. Pyridyl triazole scaffolds [73], imidazole derivatives [74] and porphyrin [75] have been involved in heterobimetallic coordination. Recently, Day et al. reported a tracer combining the naphtalimide fluorophore and a picolylamine chelator [76]. Around 55% RCY was achieved after *fac*-[^99m^Tc (CO)_3_(H_2_O)_3_]^+^ radiolabelling. The complexes showed high stability in human serum. Additionally, the rhenium(I) complexes proved to be suitable for confocal fluorescence microscopy, showing extracellular and mitochondrial uptake. However, SPECT/CT imaging revealed fast clearance (via biliary and renal pathways) and almost no uptake at the site of interest. Thus, further modifications are necessary for future imaging applications.

Dual PET/OI probes are becoming increasingly important because they allow for better spatial resolution and quantification compared to SPECT/OI. Due to the favourable nuclear physical properties and steady availability, the generator nuclide ^68^Ga is predestined for use in nuclear medicine [9]. Various chelators for gallium are suitable for multiple functionalisation. Exemplarily, dual imaging probes based on DOTA-IRDye800CW have been developed for ^68^Ga-labelling, showing that the fluorophore has no influence on the radiolabelling efficiency [77]. The promising results encouraged the study of similar systems based on ^68^Ga-NOTA [78]. In 2018, the first-in-human-PET imaging and fluorescence-guided surgery using a ^68^Ga-NOTA-IRDye800CW-bombesin were performed [79]. The novel dual probe revealed high accuracy and a strong correlation between PET and fluorescence imaging (Figure 12). This made it possible to clearly distinguish the diseased region from the healthy tissue and allowed for safe resection of glioblastoma tumours using image-guided surgery. There are other interesting chelator systems for ^68^Ga in development. Thus, Wang et al. developed an H_2_hox ligand with two 8-hydroxyquinoline moieties for ^68^Ga complexation, showing remarkable features such as: (i) easy synthesis, (ii) quantitative radiochemical yield within 5 min at room temperature and physiological pH, (iii) > 99% radiochemical purity without purification and (iv) enhanced fluorescence upon increasing gallium concentration, suitable for imaging [80].

Another study related to the use of ^68^Ga in dual-imaging probes was reported by Baranski et al. in 2018 [81]. The low molecular weight agent ^68^Ga-Glu-urea-Lys-HBED-CC was conjugated with four different fluorophores: fluorescein isothiocyanate (FITC), Alexa 488, IRDye800CW and DyLight800. All the conjugates showed high ^68^Ga complexation efficacy (RCY > 99%), indicating that the addition of the fluorophore does not affect the coordination properties of the chelator HBED. Additionally, the conjugates showed specific cell internalisation in confocal microscopy studies. Because of their NIR fluorophores, the conjugates with IRDye800CW and DyLight800 are promising for translation to the clinical area. Furthermore, the ^68^Ga-Glu-urea-Lys-HBED-CC-IRDye800CW conjugate was optimal for PSMA-specific tumour visualisation. It showed tumour enrichment and fast background clearance. Additionally, it was successfully applied for fluorescence-guided surgery in mice and pigs. This dual-imaging probe represents a promising tool for preoperative, intraoperative and postoperative detection of prostate cancer lesions. Very recently, a first-in-human-study has been reported for a patient with high-risk prostatic carcinoma [82]. The hybrid molecule PSMA-914 (^68^Ga-Glu-urea-Lys-(HE)_3_-HBED-CC-IRDye800CW) derived from PSMA-11 demonstrated its potential to accurately detect PSMA-expressing lesions before and during surgery. After 1 h post-injection, high retention of the conjugate in the tumour area was detected.

Currently, there are a number of other emerging multifunctional chelator systems with both macrocyclic and pre-organised acyclic structures, which are suitable for the development of targeted dual probes in nuclear medicine and optical imaging [83]. This means that chelator systems for further interesting PET radionuclides are also available, such as ^44^Sc, ^64^Cu and ^89^Zr.

One of the emerging radionuclides starting to be used in nuclear medicine is ^64^Cu [9]. In the development of modular dual-labelled probes, combinations of macrocyclic chelators and organic dyes dominate. Although DOTA is admittedly not the ideal chelator for Cu^II^, it is still the most widely used for dual-labelled probes with antibodies [84,85,86,87] and peptides [88,89,90]. Due to the higher stability of Cu^II^ complexes and especially the higher kinetic inertness, sarcophagins [91] and TACN [92] ligands are more suitable here. In 2014, Brand et al. reported a dual imaging probe based on sarcophagine-sulfo-Cy5 [93]. The probe was synthesised following a one-pot reaction protocol. Using carboxylic acid and amino groups of the sarcophagine cage, the ligand was equipped with a sulfo-Cy5 fluorescent tag and an exendin-4 based targeted vector for the glucagon-like peptide 1 receptor (GLP-1R). This bimodal imaging probe exhibited good performance in vivo and ex vivo for tumour imaging in mice.

TACN ligands with pyridine pendant arms form very stable Cu^II^ complexes with fast complex formation kinetics under physiological conditions [92,94]. A pyridine-bearing TACN building block with an azide group can easily be incorporated via click chemistry to a conjugate consisting of the NIR label sulfo-Cy5 and an epidermal growth factor receptor (EGFR)-targeting peptide [95]. This strategy allows for the development of targeted bimodal imaging probes based on PET (^64^Cu) and fluorescence imaging. 

Due to the more favourable complex formation kinetics compared to macrocyclic ligands, open-chain chelators for Cu^II^ are gaining in importance. These include pyridine-containing bispidine (3,7-diazabicyclo [3.3.1]nonane) ligands that are very rigid, optimally pre-organised and complementary to Cu^II^. They form Cu^II^ complexes of high thermodynamic stability and kinetic inertness very quickly under physiological conditions [96]. For more than 10 years, the potential of bispidine ligands for use in nuclear medicine has been known [97,98]. The ligand structure allows for a wide range of variations, so biological vector molecules and fluorescence tags can also be introduced [99]. However, up until today, there is only one example of bispidines used for dual imaging [100]. The reported BODIPY-bispidine probe (radiolabelled under mild conditions) displayed highly stable 6^4^Cu complexes. Despite the impact on the optical properties after ^64^Cu^II^ coordination, the decay isotopes ^64^Ni^II^ and ^64^Zn^II^ restored the quenched fluorescence. A ^64^Cu-labelled DTPA derivative with a carbo-cyanine dye (LS479, Figure 3) as a fluorescence label shows similar behaviour [101]. In addition, the C9 position of the bispidine scaffold allows for the addition of further functionalities without affecting the coordination properties. This feature can be used for the introduction of fluorescent labels and bioconjugation of targeting vectors [99]. Overall, the bispidine ligand system provides an ideal platform for the development of targeted dual imaging agents based on ^64^Cu. By increasing the denticity of bispidines, however, other interesting radionuclides for imaging and therapy such as ^111^In, ^177^Lu and ^213^Bi can also be included [102,103].

Concerning ^89^Zr complexes, which are particularly suitable for the study of longer biochemical processes, deferoxamine B (DFO) is the most used chelator and, until now, the only one studied for bimodal imaging systems. This ligand has been combined with fluorophores such as BODIPY [104], Cy5.5 [105], Cy5 [106] and more recently with IRDye 800CW [107]. Although these probes show quantitative labelling with ^89^Zr, it is known that the in vivo stability is not optimal. There are a number of DFO-based ligands, which exhibit increased stability. Among them is DFO* with additional donor groups. Comparative studies of DFO* with the gold standard DFO point to DFO* as a more suitable ligand for ^89^Zr. DFO* and its derivatives display superior stability and performance in vivo [108,109]. However, further improvements are needed, especially for solubility enhancement in the aqueous medium. So far, there have been no studies on other bimodal imaging probes based on other DFO ligands.

### 3.3. Mixed Ligand Systems

In recent years, the groups of Comba and Orvig have developed new classes of ligands by combining classical complexing agents such as pyridine, picolinate, glycinate, oxinate and phosphinate, leading to mixed ligand systems such as glycinate-oxinate, picolinate-phosphinate, oxinate-pyridine, picolinate-pyridine and bispidine-picolinate. Regarding the latter, octadentate bispidine-picolinate ligands (bispa-type) have been reported as suitable ligands for stable binding of the radiometal ions ^111^In^III^, ^177^Lu^III^ and ^225^Ac^III^ [110]. H_4_octox forms very stable complexes with ^111^In^III^ and exhibits enhanced fluorescence upon the complexation of Y^III^, Lu^III^ and La^III^ [111]. This ligand could thus be useful for non-radioactive fluorescent stability and cell studies as well as bimodal imaging. H_2_pyhox combines pyridine and oxine donor groups, resulting in an efficient chelator for ^64^Cu^II^ and ^111^In^III^. Furthermore, H_2_pyhox proved to be suitable for ^225^Ac^III^ [112]. Smaller radiometal ions such as ^44^Sc^III^, ^68^Ga^III^ and ^111^In^III^ are efficiently complexed with H_3_glyox, a ligand containing glycine and oxine donor groups [113]. This ligand shows interesting fluorescence properties after the addition of metal ions and is thus a promising system for nuclear/optical imaging [114]. H_6_dappa is a phosphinate-bearing picolinic acid-based chelating ligand for binding the radiometal ions ^111^In^III^ and ^177^Lu^III^ that has additional carboxylic acid groups for simultaneous introduction of fluorescence labels and targeting vectors [115]. H_4_pypa is a nonadentate ligand suitable for stable binding of radiometal ions such as ^111^In^III^ and ^177^Lu^III^ [116], ^44^Sc^III^ [117] and ^89^Zr^IV^ [118]. The central pyridine unit can be used in a simple way via an ether linker group to introduce targeting molecules/modules and/or fluorescence labels [117].

With the mixed ligand systems, a wide range is available for the development of customised dual imaging agents with improved complexation and pharmacokinetic properties and there are multiple possibilities for the introduction of targeting molecules/modules.

## 4. Nanoscale Systems

Nanoscale structures are categorised into nanocomposites, nanoassemblies, nanoporous and nanocrystalline materials and thus embrace organic and inorganic particles [6,119,120,121,122,123]. They differ in size, chemical composition, structure and dimension, which leads to unique properties rendering them of interest for a myriad of applications, especially in oncology. Benefitting from their large surface area to volume ratio and intrinsic properties, they serve as platforms to embed a plethora of nuclear, photoacoustic, magnetic or fluorescent modalities [120]. Many innovative nanoprobes for bi-, tri or multimodal imaging have been studied in recent decades and these are summarised elsewhere [30,36,120,124]. Great expectations are placed on these agents as they fulfil the multimodality imaging concept that achieves a more accurate diagnosis by applying just one compound. The latest advances of organic and inorganic nanoscale systems being used for nuclear and optical imaging are of relevance for this review. Either the nanoscale structures can be directly labelled or the intrinsic properties of the nanoparticles itself exhibit fluorescence for FLI, single-photon emission for SPECT or positron emission for PET. Specific targeting can be achieved by covalently attaching peptides, oligosaccharides, oligonucleotides, antibodies or immunoconjugates (e.g., for antibody based (bispecific antibodies) or cellular based immunotherapies with chimeric antigen receptor (CAR) T cells) [125,126,127]. Larger nanoscale systems (sub 100 nm), especially polymer-based nanostructures, tend to be passively accumulated in the tumour tissue through the enhanced permeability and retention effect (EPR) [128]. Besides the size, the pharmacokinetic and thus the in vivo behaviour is likewise influenced by the surface charge and shape of the nanoscale system. Crucially, the size requirement of inorganic nanoprobes was validated over the years and a value of less than 10 nm is required for them to be cleared by the renal pathway. The under-estimated impact of the protein corona formed on the surface of charged nanoscale systems influences the biodistribution, circulation and metabolism pattern [129]. Hence, the design has to be balanced carefully.

### 4.1. Organic Nanoscale Systems

Organic nanoscale systems such as liposomes [130], endosomes [131], nanocolloids [132], micelles [133] or nanocrystalline materials [106] are ideal candidates to act as drug delivery systems. However, multimodal imaging techniques are needed to track distribution in real time and quantify the accumulation of these systems in vivo. Luo et al. used common drug-loaded porphyrin-phospholipid (PoP) liposomes [134] (Figure 13). PoP itself exhibits fluorescence and is capable of chelating copper-64 used for PET. The liposomes are less than 100 nm with an excitation and emission wavelength of 675/720 nm. PET and NIR fluorescence imaging in female BALB/c mice bearing orthotopic 4T1 mammary tumours revealed a high accumulation in the liver, spleen and tumour tissue after 24 h. Tumour accumulation is attributed through the EPR effect. The quantification and reliability of FLI pinpoints the drawbacks using this single method alone. Due to the limited penetration depth of even near infrared light, in vivo fluorescence images gain limited information. The highest fluorescence intensity was observed in the tumour region, while PET biodistribution studies revealed the highest accumulation in the liver. The differences of the fluorescence signal can be explained by the various optical properties of organs and tissues.

Novel messenger vesicles which contain functional proteins and RNAs including microRNAs and mRNAs are receiving growing interest for clinical applications [135]. Those so called exosomes are considered to be non-immunogenic and non-toxic, exhibiting high stability [136]. Jung et al. monitored the biodistribution and accumulation pattern of exosomes derived from breast (murine mammary carcinoma 4T1 cell line) cancer cells in female Balb/c nu/nu mice [131]. They were functionalised with 1,4,7-triazacyclononane-triacetic acid (NOTA) to chelate the positron emitters gallium-68 and copper-64 and conjugated to the NIR infrared dye C7, exhibiting a size of approximately 100 nm. PET images revealed accumulation in the lymph nodes, liver and lung via lymphatic or hematogenous routes. In vivo fluorescence images visualised the exosomes in the brachial lymph nodes only, whereas PET images localised them also in the axillary ones. However, quantifying the fluorescence signals of ex vivo organs gave similar results to PET images.

Another bimodal nanoscale drug delivery system has been reported by Sarparanta et al. [106]. Cellulose nanocrystals (CNC) were functionalised to the chelators desferrioxamine B (DFO) or NOTA (to chelate zirconium-89 or copper-64) and the fluorescent dye Cy5. The modified CNCs exhibit diameters of less than 8 nm with an average length of 90 nm. The in vivo and ex vivo studies were examined in orthotopic 4T1 allograft-bearing mice, a tumour model of human stage IV breast cancer. The PET images clearly showed accumulation of [^64^Cu]Cu-NOTA-CNC-Cy5 and [^86^Zr]Zr-DFO-CNC-Cy5 in the liver, lung, bone and spleen. Low tumour uptake was evaluated for both materials, indicating no passive targeting through the EPR effect. Ex vivo OI images or biodistribution based on fluorescence label showed similar accumulation pattern of the compounds seen by in vivo PET imaging.

Dual-labelled dendritic polyglycerols (dPG) were equipped with camelid single-domain antibodies (sdAbs) to target the human epidermal growth factor receptor (EGFR) [137]. The dendritic polyglycerols were functionalised with a copper-64 chelator (triazacyclononane derivative) as well as with a fluorescent dye (Cy7, λ_ex_/_em_ = 750/780 nm). The hydrophobic diameter of the decorated dPG was < 8 nm. PET and OI imaging were performed in A431 tumour-bearing NMRI nu/nu mice. The PET biodistribution profile points to renal clearance and thus to a predominant renal excretion route. However, a certain accumulation of activity was found in the liver. The authors consider the non-specific binding of ^64^Cu^II^ to polyglycerol backbone as a possible reason. Tumour accumulation was low but higher in comparison to their non-targeting dPG derivatives after 24 h. Results obtained by in vivo and ex vivo OI revealed also preferred renal clearance with increased fluorescence intensity found in the kidney cortex but only minimal liver accumulation. Interestingly, the tumour uptake peaked between 24 and 48 h, which might explain the lower uptake seen in the PET image after 24 h.

One engineering approach to design organic nanoprobes with precise surface chemistry was reported by Onzen et al. [138]. Short π-conjugated oligomers self-assemble to fluorescent small molecule-based nanoparticles (SMNPs). The building blocks consisted of two fluorene units connected by a benzothiadiazole linker. Both ends contain gallic acid either decorated with alkyl or with polyethylenglycol chains exhibiting amphiphilic character. In addition, trans-cyclooctene functionalities (25%) and inert methyl (75%) groups were introduced at the periphery of the ethylene glycol chain. The SMNPs exhibited a hydrodynamic diameter of about 90 nm and the excitation and emission wavelength were 430 and 510–650 nm. To monitor the in vivo behaviour of such organic spherical nanoparticles, an ^111^In-labelled tetrazine-functionalised DOTA derivative reacted with the trans-cyclooctene unit of the oligomers in an inverse-electron-demand Diels–Alder reaction. The PET biodistribution data in nude Balb/c mice revealed significant accumulation in the liver and spleen within 70 h, with a peak at 4 h. The results demonstrated that the elimination is taking place by macrophages located in the Kupffer cells (liver) and red pulps (spleen). Due to the autofluorescence signal from the liver, in vivo imaging was not possible. Emission spectra could only be measured in blood samples.

### 4.2. Inorganic Nanoscale Systems

A plethora of inorganic-based nanoparticles such as silica [139,140,141], silicon [142,143], metal-based [144] and upconverting nanoparticles [145,146] as well as quantum dots [140,141,147] were used for PET/SPECT and optical imaging. Depending on the inorganic material, composition, size and shape, the fluorescence profile and quantum yield can vary. Semiconductor quantum dots (QDs) were considered as ideal inorganic-based alternatives to organic dyes due to their intense and narrow emission profile, higher quantum yield and their excellent photo-stability [148]. Unfortunately, they are also considered as toxic materials due to their composition of cadmium. Strategies have been designed to minimise the cytotoxicity. Cadmium telluride quantum dots were grafted on the surface of mesopourous silica (MCM-41) and radiolabelled with gallium-68 without the use of chelators [140]. The nanocomposites exhibit a size of 50 nm and were mainly accumulated in the liver, lung and kidney.

A smart quantum dot protected nanosystem was design by Shi et al. [141]. They prepared hollow mesopourous silica NPs and incorporated the commercially available QD705 (λ_ex_/_em_ = 605/700 nm) in the cavity of the HMSNs. A chimeric human/murine anti-CD105 antibody (TRC105) was grafted onto the surface of the yolk/shell-structured nanosystem which targets the membrane glycoprotein receptor CD105. It plays an important role in tumour angiogenesis, growth and metastasis. NOTA chelators (complexation of ^64^Cu) were decorated on the surface and the NPs showed a size of about 70 nm. PET images and biodistribution studies in 4T1 tumour-bearing mice revealed significant liver and spleen uptake after 24 h but also an enhanced tumour uptake in comparison to the non-targeted and blocking group. The optical imaging confirmed the results.

Intrinsically labelled zirconium-89 silica nanotags functionalised with near infrared fluorescent dyes (CF680-R, λ_ex_/_em_ = 680/700 nm) were coated with protamine and heparine to enable labelling to CAR T cells [139]. The dual-labelled nanotag gives the possibility of long-term tracking of the in vivo behaviour of such immune cells and collects information about tissue distribution by PET/FLI up to one week after adoptive cell transfer. The silica nanoparticles had a mean hydrodynamic diameter of about 120 nm. The direct CAR T cell labelling study broadcast the high silica NP loading efficiency and effective tumour uptake and demonstrated the feasibility of using these nanotags as cargos to selectively deliver drugs. Besides the application of silica NPs in medicine, silicon nanoparticles are also considered as powerful ultrasmall and non-toxic agents.

De Cola and Stephan et al. showed vividly the enormous potential of Si NPs being used as imaging agents [142,143]. In a first study, they evaluated the in vivo behaviour of Si NPs decorated with [^64^Cu]Cu-NOTA derivatives and NIR fluorescence dyes (Kodak-XS-679, λ_ex_/_em_ = 680/700 nm) [142]. In vivo imaging revealed a fast renal clearance and a significant accumulation in the liver, although a mean diameter of less than 5 nm was shown. The authors attributed this phenomenon to the difference in charge and thus the formation of the protein corona, which was not observed for neutral charged NPs. The same authors investigated the biodistribution and in vivo behaviour of dual-labelled citrate-stabilised Si NPs (<3 nm) to design neutral charged particles [143]. The NPs were functionalised with NOTA and a near infrared dye (IRDye800CW, λ_ex_/_em_ = 792/775 nm), enabling PET and OI imaging (Figure 14). It is worth noting that after the functionalisation of the dye to the Si NPs, a hypochromic shift (IR800- Si NPs, λ_ex_/_em_ = 611/753 nm) was observed, which might have been due to the presence of protonated amines. In vivo investigation by PET and OI demonstrated encouraging pharmacokinetic properties, showing quick clearance via the kidneys, no toxicity, no accumulation in organs or tissues and high stability even after excretion from the organism.

Another class of FDA-approved ultrasmall nanoparticles are AGuIX^®^ (<5 nm) where DOTAGA-Gd complexes are covalently bound on a polysiloxane matrix. Denat et al. decorated these attractive nanoparticles with NODAGA chelators (complexation of ^64^Cu and ^68^Ga) and Cy7 chromophores (IR783, λ_ex_/_em_ = 792/815 nm), enabling PET/MRI/OI trimodal imaging [149]. After functionalisation, the size of the AGuIX-NODAGA-IR753 nanoparticles increased to a mean value of about 12 nm. The in vivo evaluation in NMRI TSA tumour-bearing mice revealed excretion via the renal and hepatic pathway since significant accumulation was observed in the kidney, liver and spleen after 24 h. The authors associate the elevated uptake with the increase in the hydrodynamic diameter (>10 nm) and the application of heptamethine cyanine dyes, which has been reported to show higher hepatic uptake [150]. The OI images showed, in contrast to PET-MRI, strong fluorescence signal in the intestine and stomach and low contrast in the liver and kidney. Self-quenching effects of AGuIX-NODAGA-IR753 occurred due to the high accumulated concentrations in certain organs and tissues. The more dye absorbed in an organ or tissue, the lower the fluorescence signal is.

It is worth noting that besides the combination of fluorescence imaging with PET, Cerenkov luminescence imaging (CLI) has drawn attention in image-guided surgery, especially in combination with clinically approved radiopharmaceuticals. Cerenkov luminescence is generated due the decay processes of charged particles of sufficient energy (β-emitting nuclides). The limited application due to penetration depth and low light yield hinders further intraoperative clinical application. To extend the Cerenkov luminescence properties and enhance the signal intensity, radiolabelled NPs were considered as signal intensity enhancers and converters to achieve longer wavelength and thus deeper tissue penetration. In an exemplary study, Eu^III^-doped gadolinium oxide NPs coated with polyvinyl alcohol (PVA) for better biocompatibility were combined with ^18^F (β^+^-emitter) as an excitation source [151]. The authors demonstrated that the optical signal intensity is dependent on several factors including size/mass of NPs, surface modification, excitation distance and amount of radioactivity. Nonetheless, they proved the use of in vivo tumour NIRF imaging with high contrast and lower tissue-autofluorescence. Moreover, the intraoperative image-guided surgery successfully localised tumours and tumour boundaries.

## 5. Conclusions and Future Perspectives

It is clear that rapid advancements in the quest to find new optical imaging agents is occurring; the number of reports of new organic fluorophores and metal-based NIR imaging agents—that can be excited at higher wavelengths and that have large extinction coefficients and quantum yields and low or negligible autofluorescence properties—is increasing. This advancement is primarily due to the synthesis of organic molecules containing donor–acceptor–donor (D-A-D) moieties, enabling them to absorb in the NIR window and, in the case of lanthanide(III)-based imaging agents, when designed appropriately 2P-excitation (between 700 and 900 nm) allows for in vivo microscopy. However, it is worth noting that translation into a clinical setting is not trouble free, with further research of these imaging agents as well as translational/clinical research and regulatory affairs required in order for them to reach their potential as highly active theranostic agents. The approval process for diagnostic tracers is very similar to that observed in traditional drug discovery, that is, after their discovery and testing, pre-clinical trials followed by government approval is required.

Metal complexes that have been labelled with both a fluorophore and radionuclide is an exciting area that is in its infancy. Within this area is the ability for the ligand to be labelled with a non-radionuclide lanthanide(III) ion, for 2P-excitation, or with a radionuclide for radionuclide scanning. Co-dosed, it is expected that these two imaging agents, due to almost identical properties, would locate to the same site of action and thus dual imaging with increased spatial resolution could be accomplished.

An alternate area is dual imaging probes that contain both fluorophores and radionuclides. There has been rapid development in this field in recent years. This includes small molecule probes, new modular and nanoscale systems with improved detection and pharmacokinetic properties. Especially in the last 5–10 years, a number of new BFCAs with improved complexation properties have been developed that can be used for classical radiometals such as technetium-99m and indium-111. They also provide access to emerging radiometals such as scandium-44, copper-64, gallium-68, zirconium-89 and lutetium-177. A number of new conjugation strategies are available that allow for the introduction of targeting modules under mild conditions. These include methods of bioorthogonal chemistry, such as the Staudinger–Bertozzi ligation, the strain-promoted alkyne-azide cycloaddition and the inverse electron demand Diels–Alder reaction. Furthermore, enzyme-mediated conjugation strategies are increasingly used to achieve a defined functionalisation of, for example, proteins (antibodies and their fragments) and nucleic acids. This provides a wide range of new tools for personalised medicine and precision surgery.

Whilst in its infancy, novel imaging agents comprised of nanoscale systems are being approved for use in the clinic [152]. The approval of imaging agents containing nanoscale systems does have its challenges, with reproducibility of the manufactured systems being critical [153,154,155,156]. In response to this, the FDA has released a number of guidance documents to provide information to academics and industry on the development, manufacturing and use of some products that contain nanomaterials [157]. The powerfulness of science is always evolving, and the chemistry, analytical systems for physio-chemical characterisation and policies are being created to allow for the safe use of novel dual-labelled nanoscale systems.

Image-guided surgery, including robot-assisted surgery, is particularly attractive for improving prospects of curing, especially in the field of oncology. Fluorescence-guided surgery is the logical evolution of radio-guided surgery, because it allows for detailed real-time visualisation, enabling surgical removal of all diseased tissue during an operation [158,159,160]. However, the sole use of optical imaging, especially in humans, is limited to regions close to the surface due to the rather low penetration depth of the light radiation. For this reason, dual-labelled probes (nuclear and fluorescence) are increasingly used for deeper regions. This makes it possible to detect the diseased regions externally by means of nuclear imaging and subsequently to clearly distinguish the stained diseased tissue from the healthy area internally. Prominent examples of the use of such hybrid probes can be found in the fields of sentinel lymph node biopsy [161,162,163,164,165,166,167], prostate cancer [168,169], neuroendocrine tumours [170] and breast [171] and kidney cancer [172].

Cerenkov emission is a method that does not require additional fluorophores and is under discussion for clinical application [173,174,175]. However, depending on the Cerenkov intensity of the radionuclides used, the signal intensity is three to four orders of magnitude lower compared to fluorescence-emitting probes [176]. In order to keep the radiation dose for patients and clinical staff low, the activity concentration of the radionuclides applied must be kept as low as possible. This limits the application possibilities, especially for Cerenkov-emitting nuclides. For a clinical application of Cerenkov imaging, the detection sensitivity must be significantly increased.

In terms of image-guided surgery, dual-labelled (nuclear and fluorescence) probes will dominate this exciting field in the coming years and open up new fields of application. Here, methods of artificial intelligence will also increasingly be incorporated [177]. With regard to clinical application, various challenges have to be overcome. This concerns, for example, the production of ready-to-use kits that have sufficient long-term stability. The translation of suitable dual-labelled imaging probes into clinical routine requires regulatory approval and, in turn, manufacturing under the conditions of good manufacturing practice (GMP). Overall, this is a challenging and exciting field that requires intensive multidisciplinary collaboration between experts in different fields and will undoubtedly lead to new products that enable improved non-invasive imaging with more sophisticated treatment options.

## Figures and Tables

**Figure 1 pharmaceuticals-15-00432-f001:**
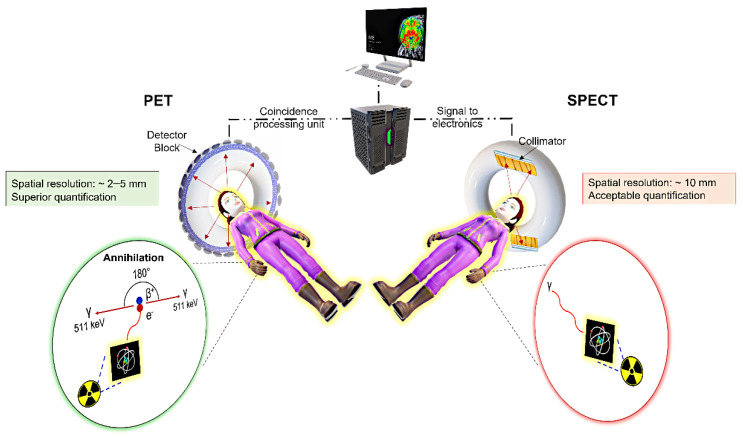
Depiction of PET and SPECT imaging.

**Figure 2 pharmaceuticals-15-00432-f002:**
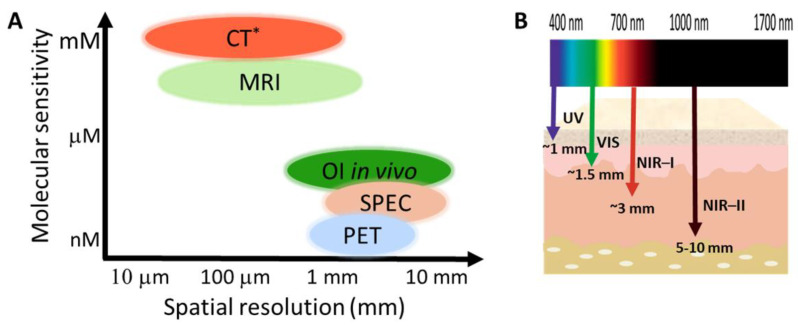
(**A**) The molecular sensitivity and spatial resolution of imaging processes of relevance to this review, * sensitivity not well characterised. Data obtained from [17,18,19]. (**B**) Schematic of penetration depth at varying wavelengths.

**Figure 3 pharmaceuticals-15-00432-f003:**
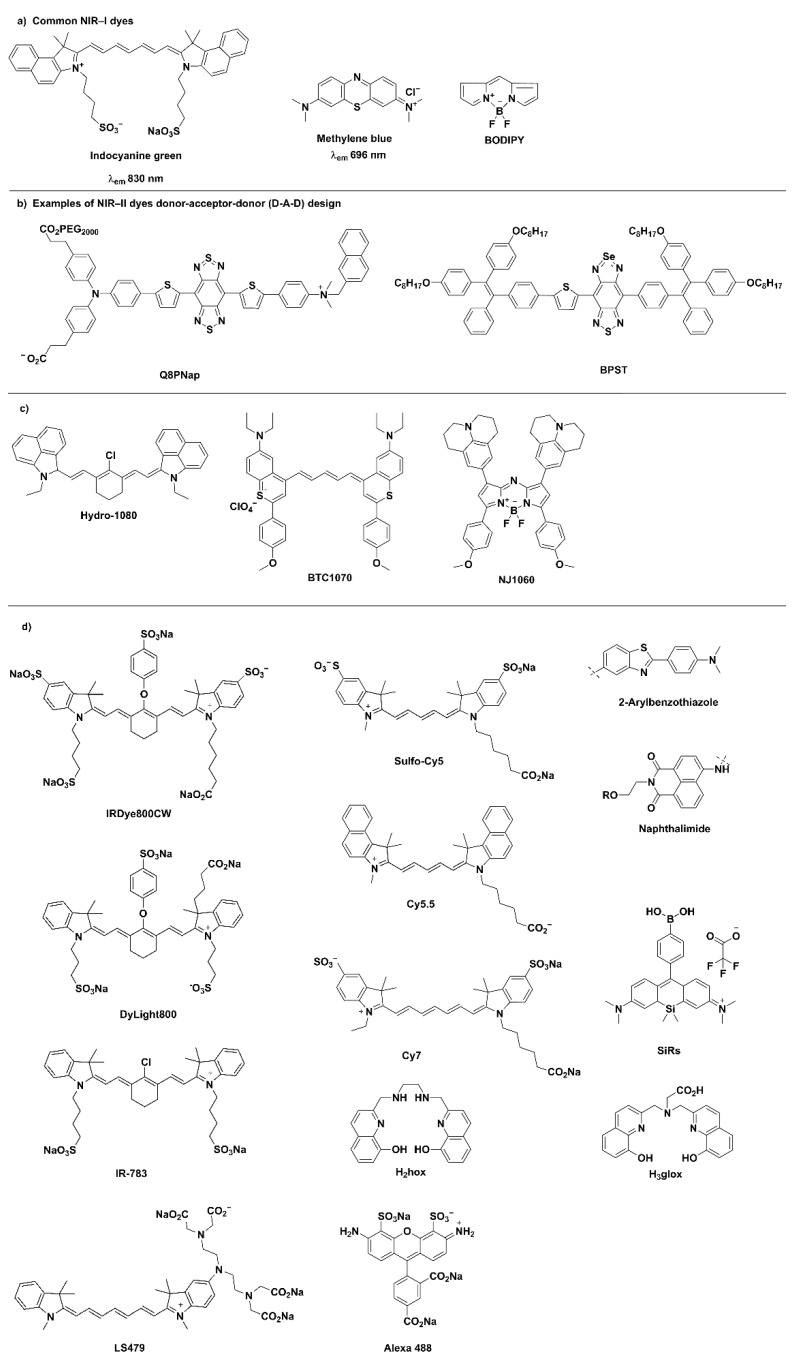
Representative examples of (**a**) Common NIR-I dyes, (**b**) NIR-II dyes that are based on the donor–acceptor–donor design, Q8PNap [25] and BPST [26], (**c**) Alternate NIR-II dyes: Hydro-1080 [27], BTC1070 [28] and NJ1060 [29] and (**d**) Additional fluorophores, not previously noted, that are relevant to Section 3 and Section 4.

**Figure 4 pharmaceuticals-15-00432-f004:**
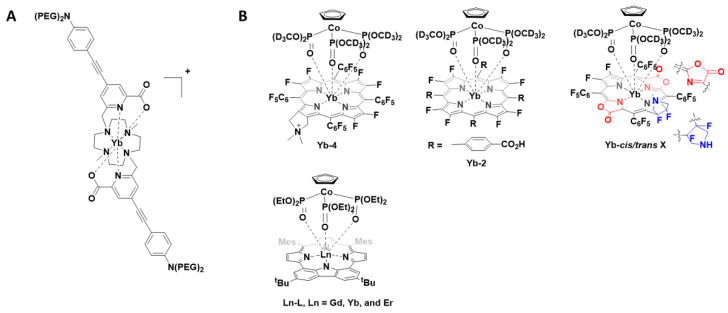
(**A**) Chemical structure of the -coordinate cationic [YbL]^+^ complex; (**B**) Structures of porphyrinates complexes Yb-4, Yb-2, Yb-*cis*/*trans*-3 and Ln-L.

**Figure 5 pharmaceuticals-15-00432-f005:**
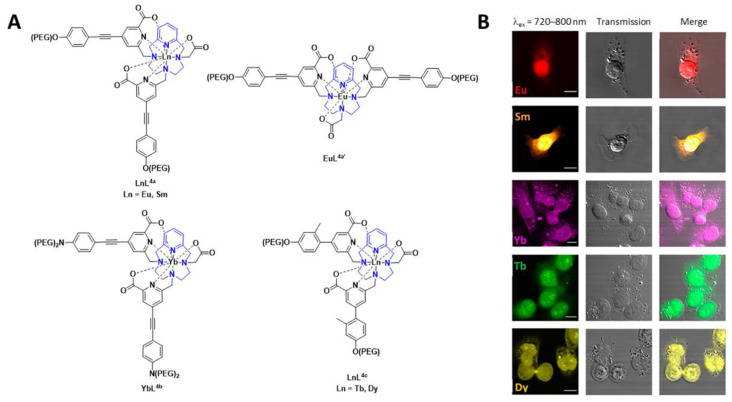
(**A**) Chemical structures of lanthanide-based probes containing pyclen ligands. (**B**) Left: 2P imaging of paraformaldehyde-fixed T24-cells: [EuL^4a^], [SmL^4a^] (λ_ex_ = 750 nm), [YbL^4b^] (λ_ex_ = 800 nm), [TbL^4c^] and [DyL^4c^] (λ_ex_ = 720 nm). Middle: transmitted light DIC images. Right: merged images. Reprinted (adapted) with permission from Ref. [52]. Copyright 2020 American Chemical Society.

**Figure 6 pharmaceuticals-15-00432-f006:**
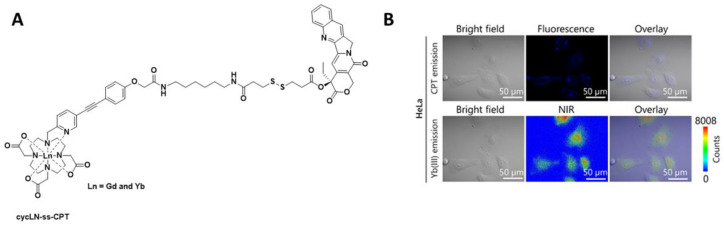
(**A**) Chemical structure of cycLN-ss-CPT. (**B**) NIR imaging of YbNPs in HeLa cells; CPT emission λ_ex_ 370 nm and λ_em_ 400–450 nm; Yb(III) emission λ_ex_ 380 nm and λ_em_ 900–1700 nm. Reprinted (adapted) with permission from Ref. [54]. Copyright 2021 American Chemical Society.

**Figure 7 pharmaceuticals-15-00432-f007:**
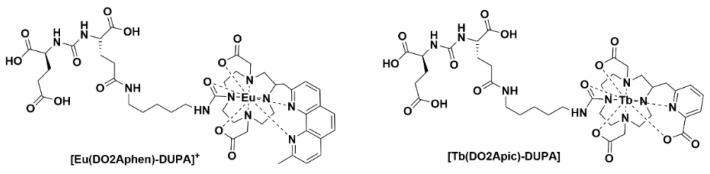
Chemical structures of [Eu(DO2Aphen)-DUPA]^+^ and [Tb(DO2Apic)-DUPA].

**Figure 8 pharmaceuticals-15-00432-f008:**
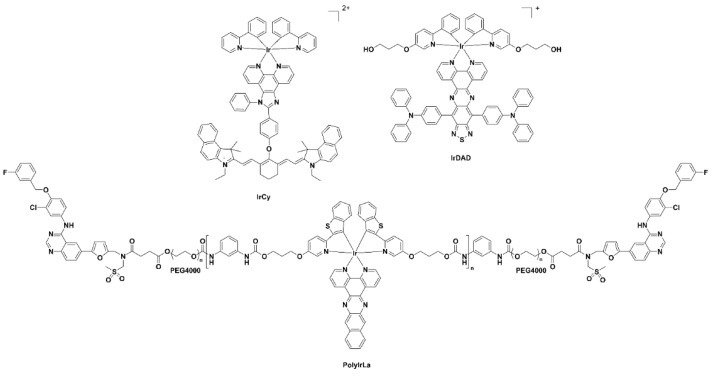
Chemical structures of the iridium(III) complexes IRCY, IRDAD and PolyIRLA.

**Figure 9 pharmaceuticals-15-00432-f009:**
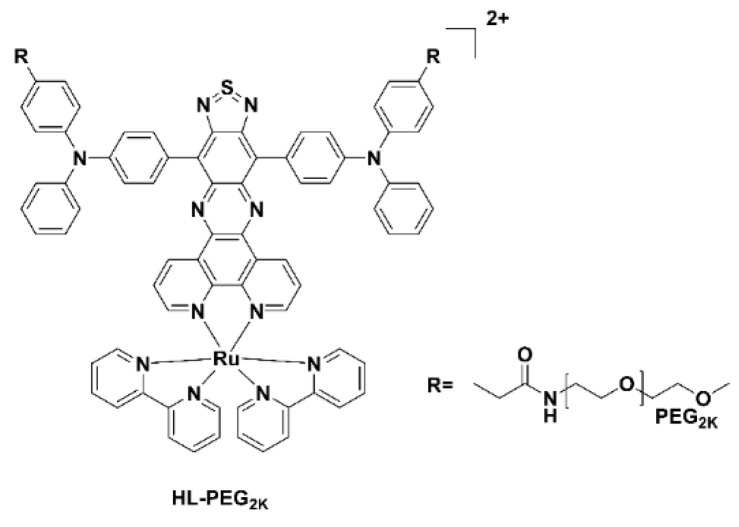
Chemical structure of the Ru(II) polypyridyl complex, HL-PEG_2K_.

**Figure 10 pharmaceuticals-15-00432-f010:**
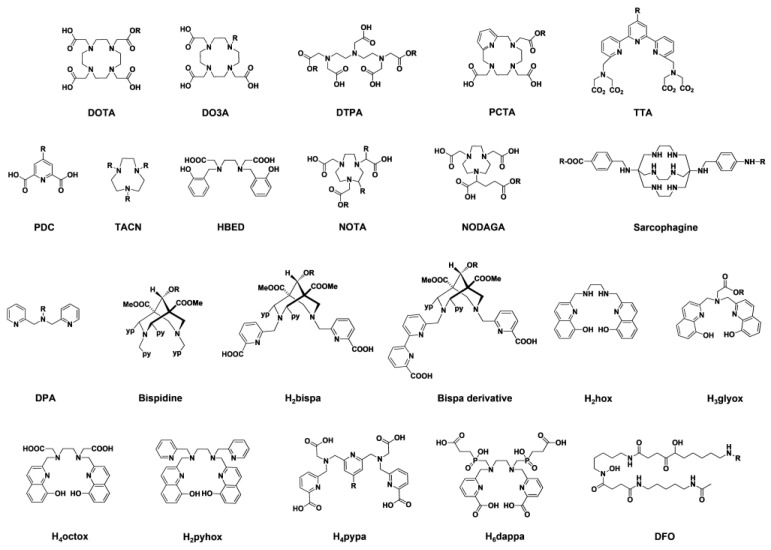
Representative chelators discussed in this review.

**Figure 11 pharmaceuticals-15-00432-f011:**
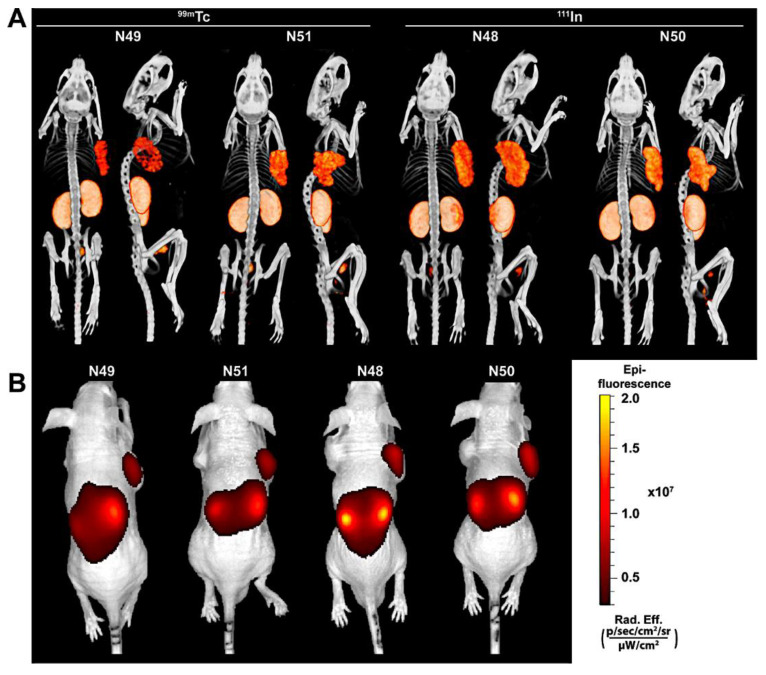
The ligands visualise PSMA-positive tumours using µSPECT/CT and fluorescence imaging. The panels display (**A**) µSPECT/CT scans, (**B**) Fluorescence images of mice with LS174T-PSMA (right) and wild-type LS174T (left) tumours after intravenous injection of ^111^In (10 MBq/mouse) or ^99m^Tc-labelled (15 MBq/mouse) ligands. Reprinted with permission from Ref. [71]. Copyright 2022 American Chemical Society.

**Figure 12 pharmaceuticals-15-00432-f012:**
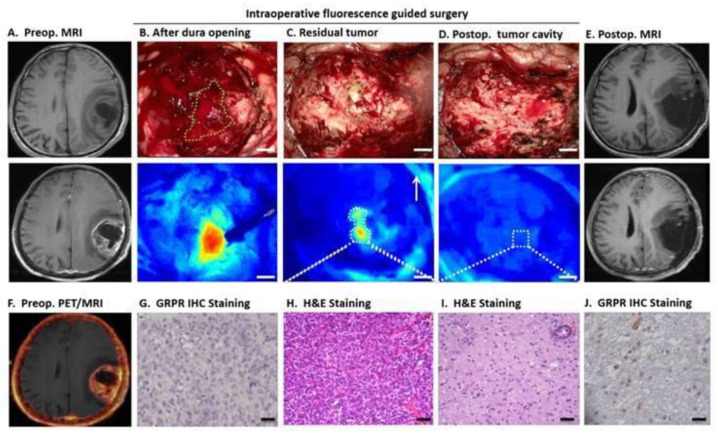
Dual-modal imaging for glioblastoma resection using ^68^Ga-NOTA-IRDye800CW-bombesin. The glioblastoma tumour is visible by MRI after gadolinium injection. (**A**) Uptake in the tumour area was observed by PET/MRI, (**F**) The NIRF assessment displayed the fluorescent tumour ((**B**): scale bar 5 mm). (**C**) Displays the fluorescent residual tumour tissue, and (**H**) the histological staining (scale bar 50 µm). Several antigen-expressing cells were found in the tissue after gastrin-releasing peptide staining ((**G**): scale bar 20 µm). After resection, no residual fluorescence was detected in the tumour’s cavity. (**D**) Moreover, the tissue along the cavity was confirmed to be normal (**I**) and few antigen-expressing cells were found on it (**J**). The post-operative analysis shows total removal of the malignancy (**E**). Reprinted with permission from Ref. [79]. Copyright 2018 Ivyspring International Publisher.

**Figure 13 pharmaceuticals-15-00432-f013:**
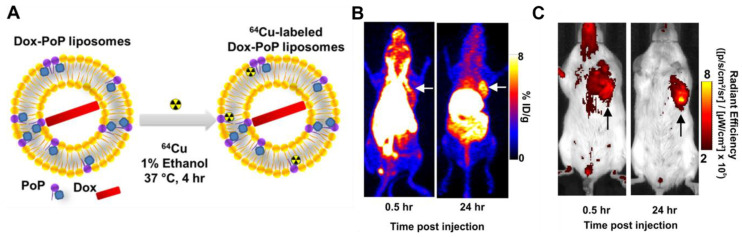
(**A**) Doxorubicin-loaded Cu-porphyrin-phospholipid liposomes. (**B**) PET scan of mice after administration of DOX-CuPoP. (**C**) NIR Fluorescence imaging of mice after administration of DOX-CuPoP. Reprinted with permission from [134]. Copyright 2017 American Chemical Society.

**Figure 14 pharmaceuticals-15-00432-f014:**
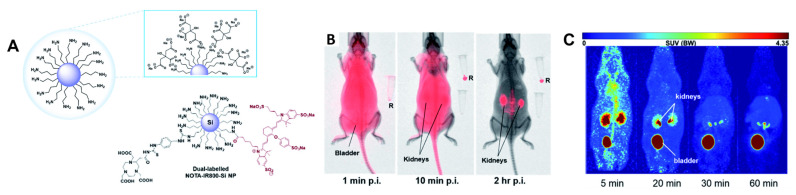
(**A**) Dual-labelled NOTA-IRDye800CW-Si nanoparticle coated with citric acid. (**B**) In vivo optical imaging scans of healthy nu/nu mice intravenously injected with IRDye800CW-Si NPs after 1 min, 10 min and 2 h p.i. (R: injected IR800-Si NPs as reference). (**C**) In vivo PET imaging of [^64^Cu]Cu-NOTA-IRDye800CW-Si NPs 5, 20, 30, 60 min p.i. Reprinted with permission from [143]. Copyright 2020 The Royal Society of Chemistry.

## Data Availability

Data sharing not applicable.

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
