# Peer review of "Dual-Labelling Strategies for Nuclear and Fluorescence Molecular Imaging: Current Status and Future Perspectives"

_pharmaceuticals, 2022, doi:10.3390/ph15040432_

Round 1
Reviewer 1 Report
Authors reported an updated summary of recent development in dual modality imaging agents. It is definitely timely. The manuscript was well-organized and well-written. However, some issues need to be addressed before its publication.
- Throughout the text: There is no description on the criteria of an ideal imaging agent for dual modality imaging. What is needed for a new agent to become a diagnostic tracer in clinical practice? Does any agent presented in this report have the potential for widespread clinical applications? This must be addressed before its publication.
- Throughout the text: There is no description on pharmacokinetic requirements of an dual-modality imaging agent. The whole paper looks like a compilation of literature data. Authors should provide their own teaching in terms of agent development (as a clinically useful drug not another new identity).
- Throughout the text: Authors need to pay attention to isomerism of metal chelates. Since metal chelates are often relatively large, the presence of isomerism may pose a significant challenge with respect to receptor binding affinity and pharmacokinetic properties of the resulting imaging agent.
- Figure 12: Should it be "Bombesin"?
- Nanoscale system: Do authors know the exact composition of those nanoscale agent(s) for dual modality imaging? Are you sure if FDA will approve anything for injection into human subjects without knowing the exact composition or chemical form of the agent?
- What is the shortcoming for the reported dual modality imaging agents? This will have a significant impact on future development in this area.
Author Response
Reviewer 1:
Authors reported an updated summary of recent development in dual modality imaging agents. It is definitely timely. The manuscript was well-organized and well-written. However, some issues need to be addressed before its publication.
Many thanks for the overall positive comments on our review.
- Throughout the text: There is no description on the criteria of an ideal imaging agent for dual modality imaging. What is needed for a new agent to become a diagnostic tracer in clinical practice? Does any agent presented in this report have the potential for widespread clinical applications? This must be addressed before its publication.
The introduction of diagnostic tracers into the clinic is not really the focus of this review. However, we agree with the reviewer that this is an important aspect. Accordingly, we have introduced the following text in conclusions:
“The approval process for diagnostic tracers is very similar to that observed in traditional drug discovery, that is, after their discovery and testing, pre-clinical trials followed by government approval is required.”
- Throughout the text: There is no description on pharmacokinetic requirements of an dual-modality imaging agent. The whole paper looks like a compilation of literature data. Authors should provide their own teaching in terms of agent development (as a clinically useful drug not another new identity).
Naturally, the pharmacokinetics of the imaging agent will depend on the agent itself and the mode of delivery. We agree with this reviewer that we have not mentioned the pharmacokinetic requirements. The text below addresses this and it was inserted in the introduction:
“The pharmacokinetic requirements of dual-labelled imaging agents in vivo is dependent on the mode of delivery as well as timeframe between administration and imaging test. Furthermore, the type of chelators, dye molecules and target modules, as well as their chemical linkage, will vary these properties and so comprehensive pre-clinical studies are needed for further developments. Some of the examples discussed below, utilise ligands or fluorophores that have been approved for use in the clinic. However, a comprehensive understanding of the agent’s absorption, distribution, metabolism, and excretion (ADME) properties, as well as toxicity, is naturally needed before phase III trials can commence.“
- Throughout the text: Authors need to pay attention to isomerism of metal chelates. Since metal chelates are often relatively large, the presence of isomerism may pose a significant challenge with respect to receptor binding affinity and pharmacokinetic properties of the resulting imaging agent.
We are not sure which section the reviewer is referring to. However, all structures have been drawn and as per the original paper, and the main body text also describes the findings and conclusions within that work.
- Figure 12: Should it be "Bombesin"?
This change has been made.
- Nanoscale system: Do authors know the exact composition of those nanoscale agent(s) for dual modality imaging? Are you sure if FDA will approve anything for injection into human subjects without knowing the exact composition or chemical form of the agent?
We agree that the translational factor is a crucial step in developing nanoscale systems. However, the authors focused on the progress of dual-labelled nanoscale systems within the last five years. Not many of them discussed herein (such as AGuIX NPs) are clinical-approved nanomaterials and have been used for dual-multimodal imaging. For over a decade the FDA has been investing in the field of nanotechnology, with the research performed at National Centre for Toxicological Research (NCTR), allowing a better understanding of possible concerns associated with their use. For any approval by the FDA, stringent quality control, pre-clinical, and clinical studies are needed; there is considerable investment by the FDA in the development of tools and approaches to ensure that those that will be approved are safe. To date there are a number of nanoparticle systems that have been approved for use and many others that are in pre-clinical trials. Recently published reviews point to the challenges of clinical translation:
Preclinical Cancer Theranostics—From Nanomaterials to Clinic: The Missing Link, doi.org/10.1002/adfm.202104199
The Clinical Translation of Organic Nanomaterials for Cancer Therapy: A Focus on Polymeric Nanoparticles, Micelles, Liposomes and Exosomes, doi.org/10.2174/0929867324666170830113755
Inorganic nanoparticles in clinical trials and translations, doi.org/10.1016/j.nantod.2020.100972
Inorganic hybrid nanoparticles in cancer theranostics: understanding their combinations for better clinical translation, doi.org/10.1016/j.mtchem.2020.100381
Nanoparticles in the clinic: An update, doi.org/10.1002/btm2.10143
To address these comments, we have added the following text and references in the conclusion section:
“Whilst in its infancy, novel imaging agents comprising of nanoscale systems are being approved for use in the clinic. The approval of imaging agents, containing nanoscale systems does have its challenges, with reproducibility of manufactured systems being critical. In response to this, the FDA has released a number of guidance documents to providing information to academics and industry on the development, manufacturing and use of some products that contain nanomaterials. (https://www.fda.gov/science-research/nanotechnology-programs-fda/nanotechnology-guidance-documents) The powerfulness of science is always evolving, and the chemistry, analytical systems for physio-chemical characterisation, as well as policies, are being created to allow the safe use of novel dual-labelled nanoscale systems.”
- What is the shortcoming for the reported dual modality imaging agents? This will have a significant impact on future development in this area.
We believe that this has addressed in the conclusion to the manuscript. This review summarises the recent findings and highlights their clinical potential, with a positive outlook on the use and application of these dual-modality imaging agents, rather than their shortcomings.
Reviewer 2 Report
Review MDPI pharmaceuticals: Dual-Labelling Strategies for Nuclear and Fluorescence Molecular Imaging: Current Status and Future Perspectives by Manja Kubeil et al.
This paper reviews the developments and state-of-the-art of dual labeling strategies for nuclear and fluorescence imaging. The review is a robust overview, although limited to imaging of cancer only. The authors sum up the different radioisotopes and fluorophores in various combinations. It would be nice to show a table including the pros and cons of the multiple labels (OI and radioactivity) and why they are used in different settings. Furthermore, such a table should include eventual clinical use. Overall, the translational aspect in this review is limited. For instance, the clinical use of fluorophores is limited, and their toxicity profiles were not discussed. The study will have a higher impact if the aspect best-of-both-worlds is explored more intensively on their potential clinical use.
Designing drugs would be possibly leading to new diagnostics in medicine. The aspects such as production costs, availability, waste management of the isotopes, etc., are warranted.
From our experience, various chelators, dyes, and their positioning on the targeting molecule will impact their biodistribution, clearance, and target (receptor) binding site. This should be discussed as well.
Author Response
Reviewer 2:
Comments and Suggestions for Authors
Review MDPI pharmaceuticals: Dual-Labelling Strategies for Nuclear and Fluorescence Molecular Imaging: Current Status and Future Perspectives by Manja Kubeil et al.
This paper reviews the developments and state-of-the-art of dual labeling strategies for nuclear and fluorescence imaging. The review is a robust overview, although limited to imaging of cancer only.
Many thanks for the overall positive comment on our review.
The authors sum up the different radioisotopes and fluorophores in various combinations. It would be nice to show a table including the pros and cons of the multiple labels (OI and radioactivity) and why they are used in different settings. Furthermore, such a table should include eventual clinical use.
The review has focused only on cancer because otherwise it would have been too long and the clarity would have been lost. A comprehensive compilation of nuclear/fluorescence imaging molecular and nanoscale tools as well as the corresponding pros and cons is provided in the very recent review of Ariztia et al. in Bioconjugate Chemistry 2022, 33, 24-52. We have referred to this in the introduction and included the following text:
“In the latter review, an extensive compilation of nuclear and fluorescence imaging molecular and nanoscale tools with a discussion of pros and cons for special biological applications can be found.”
Overall, the translational aspect in this review is limited. For instance, the clinical use of fluorophores is limited, and their toxicity profiles were not discussed. The study will have a higher impact if the aspect best-of-both-worlds is explored more intensively on their potential clinical use.
This review has summarised and critically discussed the state of the art of research, especially in the last 5 years. We are aware that we have not addressed all aspects, but we think that in particular the chapter conclusions and future directions provides enough suggestions to move forward in the development of novel dual-labelled probes that combine the best-of-both-worlds. For this reason, we have also provided novel chelators and fluorescent metal complexes with improved properties in the preceding chapters. Ultimately, the application depends on the specific target and accordingly always requires special optimisation steps.
Clinical use is certainly the ultimate goal of the development of new dual-labelled probes. However, this is a long way to go and goes beyond the scope of this review. This is shown, for example, by the indocyanine green (ICG) dye, which was already approved by the FDA in 1959. Even though there are dye molecules with better photo-physical properties today, clinical approval is still pending.
Concerning the toxicity profiles, the amounts of fluorophores used and particularly of radiotracers are typically much lower than toxic doses.
Designing drugs would be possibly leading to new diagnostics in medicine. The aspects such as production costs, availability, waste management of the isotopes, etc., are warranted.
We agree with the reviewer and think this is summed up in conclusions.
From our experience, various chelators, dyes, and their positioning on the targeting molecule will impact their biodistribution, clearance, and target (receptor) binding site. This should be discussed as well.
We fully agree with the reviewer. Accordingly, the following text has been added to the introduction:
“The pharmacokinetic requirements of dual-labelled imaging agents in vivo is dependent on the mode of delivery as well as timeframe between administration and imaging test. Furthermore, the type of chelators, dye molecules and target modules used, as well as their chemical linkage, will vary these properties and so comprehensive pre-clinical studies are needed for further developments. Some of the examples discussed below, utilise ligands or fluorophores that have been approved for use in the clinic. However, a comprehensive understanding of the agent’s absorption, distribution, metabolism, and excretion (ADME) properties, as well as toxicity, is naturally needed before phase III trials can commence.“